# Long Non-Coding RNAs in Diffuse Large B-Cell Lymphoma

**DOI:** 10.3390/ncrna7010001

**Published:** 2020-12-28

**Authors:** Kasper Thystrup Karstensen, Aleks Schein, Andreas Petri, Martin Bøgsted, Karen Dybkær, Shizuka Uchida, Sakari Kauppinen

**Affiliations:** 1Center for RNA Medicine, Department of Clinical Medicine, Aalborg University, DK-2450 Copenhagen, Denmark; kathka@dcm.aau.dk (K.T.K.); schein.aleks@gmail.com (A.S.); andreas.petri@gmail.com (A.P.); 2Department of Clinical Medicine, Faculty of Medicine, Aalborg University, DK-9000 Aalborg, Denmark; martin.boegsted@rn.dk (M.B.); k.dybkaer@rn.dk (K.D.); 3Department of Haematology, Clinical Cancer Research Center, Aalborg University Hospital, DK-9000 Aalborg, Denmark

**Keywords:** DLBCL, lncRNA, miRNA, rituximab

## Abstract

Diffuse large B-cell lymphoma (DLBCL) is the most common lymphoid malignancy in adults. Although significant progress has been made in recent years to treat DLBCL patients, 30%–40% of the patients eventually relapse or are refractory to first line treatment, calling for better therapeutic strategies for DLBCL. Long non-coding RNAs (lncRNAs) have emerged as a highly diverse group of non-protein coding transcripts with intriguing molecular functions in human disease, including cancer. Here, we review the current understanding of lncRNAs in the pathogenesis and progression of DLBCL to provide an overview of the field. As the current knowledge of lncRNAs in DLBCL is still in its infancy, we provide molecular signatures of lncRNAs in DLBCL cell lines to assist further lncRNA research in DLBCL.

## 1. Introduction

Recent data implies that at least 80% of the human genome is transcribed, producing about 20,000 messenger RNAs (mRNAs), over 2300 microRNAs (miRNAs), a set of structural RNAs (such as ribosomal RNAs (rRNAs), transfer RNAs (tRNAs), and small nuclear RNAs (snRNAs)), and at least 50,000 long noncoding RNAs (lncRNAs) [1,2,3]. LncRNAs comprise a diverse group of non-protein coding transcripts that are longer than 200 nucleotides in length, lack an open reading frame, and regulate gene expression at many levels by affecting transcription, cellular localization, mRNA stability, translation, and other post-transcriptional events [4,5,6,7]. As a result, lncRNAs regulate a broad range of biological processes, including aging, cell growth and differentiation, hematopoiesis, and immune response [8,9,10,11]. Accordingly, numerous studies have reported highly diverse roles for lncRNAs in cancer [12,13,14], including development and maintenance of resistance to anti-cancer therapy [15,16,17,18]. Most prominent examples involve well-characterized lncRNAs, such as CDKN2B antisense RNA 1 (*CDKN2B-AS1*, also known as *ANRIL*) [19], growth arrest specific 5 (*GAS5*) [20,21], H19 imprinted maternally expressed transcript (*H19*) [22,23], HOX transcript antisense RNA (*HOTAIR*) [24,25], metastasis associated lung adenocarcinoma transcript 1 (*MALAT1*) [26,27], and X inactive specific transcript (*XIST*) [28,29]. However, the functions of lncRNAs in the pathogenesis of B-cell lymphomas are still poorly understood.

Lymphomas are a collective term used for malignancies of the lymph system, and they are divided into Hodgkin’s lymphomas (10% of lymphomas) and non-Hodgkin lymphomas (NHL), which account for ca. 90% of all lymphomas [30,31,32]. NHLs comprise a diverse group of lymphoid cancers with approx. 85%–90% of NHL cases originating from B cells, while the remaining NHLs derive from T cells or NK cells [33,34]. Diffuse large B-cell lymphoma (DLBCL) is the most common form of NHL, accounting for up to 40% of all lymphomas, with an incidence rate of approximately five to six cases per 100,000 Americans per year [35]. The 5-year survival rate ranges from 50%–80%, depending on risk profile, and the first-line treatment is the R-CHOP regimen (Rituximab plus Cyclophosphamide, Hydroxydaunorubicin, Oncovin, Prednisone combination therapy) [36]. Rituximab, a chimeric anti-CD20 monoclonal antibody [37] was added to the standard CHOP chemotherapy regimen in 2003, which improved the two-year progression-free survival from 51% to 61% and the overall survival rate from 52% to 78% [38,39]. Nevertheless, 30%–40% of the patients eventually relapse or are refractory to R-CHOP, experiencing poor two-year outcomes with overall survival rates of only 20%–40% [40,41], which calls for improved molecular understanding of the disease and new therapeutic strategies for the treatment of DLBCL. To this end, lncRNAs might be a key to further understand the diagnostic and prognostic measures for DLBCL to promote better treatment options for DLBCL patients. In this article, we summarize the current status of lncRNAs in DLBCL and provide a survey for lncRNAs in DLBCL cell lines to assist further research into the biological functions of lncRNAs in the pathogenesis of DLBCL.

## 2. General Functions of lncRNAs

LncRNAs exert their functions by binding to other macromolecules: DNA, RNA, or proteins. At the level of transcription, a lncRNA can form a scaffold for the epigenetic or transcription factor complex to assist its activation or repression of specific set of genes. The best investigated example is the binding between histone H3 lysine 27 methyltransferase, polycomb repressive complex 2 (PRC2) (i.e., the functional enzymatic component, enhancer of zeste homolog 2 (EZH2)) and lncRNAs such as *HOTAIR* [42], taurine up-regulated 1 (*TUG1*) [43], and *XIST* [44]. However, recent studies suggest that EZH2 binds many RNAs promiscuously, including lncRNAs [45,46,47,48,49,50]. Besides being transcriptional regulators, some lncRNAs can bind other RNAs to exert post-transcriptional regulation. For example, lncRNAs have been suggested to function as competitive endogenous RNAs (ceRNAs) to sequester miRNAs thereby influencing the amount of proteins being produced from the target mRNAs. Such lncRNAs are more commonly known as miRNA sponges, which include the *H19* lncRNA sponging *miR-675* [51], the hepatocellular carcinoma up-regulated long non-coding RNA (*HULC*) binding to *miR-372* [52], and the ZNFX1 antisense RNA 1 (*ZFAS1*) sequestering *miR-150* [53]. As for the translational control, lncRNAs can bind to RNA-binding proteins, which regulate mRNA translation. For example, in murine cardiomyocytes, the paternally imprinted lncRNA gene, antisense of IGF2R non-protein coding RNA (*Airn*, also known as *Air*), binds to the RNA-binding protein insulin like growth factor 2 mRNA binding protein 2 (Igf2bp2) to regulate the translational efficiency of hundreds of mRNAs to control the homeostasis of cardiomyocytes. In addition, lncRNA sequences themselves may be important for genomic imprinting, an epigenetic phenomenon through which genes are expressed in a parent-of-origin dependent manner, as in the case of the paternal imprinting lncRNA *Airn* [54] and the maternal imprinting lncRNA *H19* [55]). Genomic lncRNA sequences may also contain important regulatory sequences, such as enhancers, as in the case of Hand2, opposite strand 1 (*Hand2os1*, also known as *Upperhand* [56]) and Hand2 downstream lncRNA (*Hdnr*, also known as *Handsdown* [57])].

Because of the broad definition of lncRNAs as non-protein-coding transcripts longer than 200 nucleotides in length, pseudogenes, which arise from the duplication of DNA sequences or retrotransposition and integration into the genome [58], are currently considered as a new category of lncRNAs [59]. Furthermore, circular RNAs (circRNAs) have emerged as an abundant class of covalently closed non-coding RNA molecules in eukaryotes that arise from backsplicing of protein-coding and non-coding exons and/or introns [60,61,62]. Due to the lack of free 5′ and 3′ ends, circRNAs are more resistant to exonuclease-mediated degradation compared to linear RNAs (e.g., mRNAs). Because of their stability, circRNAs accumulate in the blood, saliva, and urine, which have made them attractive candidates for biomarker discovery [63,64]. Finally, some lncRNAs have been shown to encode small peptides, named micropeptides. The examples of micropeptides are myomixer, myoblast fusion factor (*MYMX*) [65], and myoregulin (*MRLN*) [66] which regulate muscle formation and activity, respectively. Taken together, lncRNAs comprise a heterogeneous class of non-protein coding transcripts with highly diverse roles in biological processes. However, additional functional studies are required to establish the functions of lncRNAs in human diseases, such as cancer.

## 3. LncRNAs in DLBCL

Due to the increased interest to study lncRNAs in recent years, researchers have also identified a number of differentially expressed lncRNAs in DLBCL patients compared to healthy donors or in DLBCL cell lines [67,68,69,70]. For example, the p53-activated lncRNA, promoter of CDKN1A antisense DNA damage activated RNA (*PANDA*), was reported to inactivate the MAPK/ERK signaling pathway to suppress cell growth by a G0/G1 cell cycle arrest [68] (Figure 1A). By comparison, the firre intergenic repeating RNA element (*FIRRE*) lncRNA was shown to function as an oncogene by activating Wnt/β-catenin signaling pathway via promoting nuclear translocation of β-catenin [71] (Figure 1B). Other lncRNAs, acting as post-transcriptional regulators, include the DBH antisense RNA 1 (*DBH-AS1*) binding to the RNA-binding protein, BUD13 homolog (BUD13), controlling the translation of fibronectin 1 (*FN1*) to positively regulate the proliferation, migration, and invasion of DLBCL cells [72] (Figure 1C). Another example is the EZH2-mediated lncRNA, FAS antisense RNA 1 (*FAS-AS1*), which binds to the RNA-binding protein, RNA binding motif protein 5 (RBM5), to control alternative splicing of Fas cell surface death receptor (*FAS*) mRNA [73]. Yet, other lncRNAs were shown to directly control protein modifications to influence tumor growth, as in the case of *TUG1*, which reduces protein levels of the MET proto-oncogene, receptor tyrosine kinase (MET) by promoting its ubiquitination [74].

Among various functions of lncRNAs, there is an increasing trend to investigate lncRNAs as miRNA sponges by viewing lncRNAs as an extra layer of the post-transcriptional regulatory machinery to fine-tune miRNA-mediated control of protein abundance. For example, the nuclear paraspeckle assembly transcript 1 (*NEAT1*) was shown to bind *miR-34b-5p* to affect the proliferation of DLBCL cell lines by targeting the GLI family zinc finger 1 (*GLI1*) [75]. Two lncRNAs have been implicated in the proliferation of DLBCL cells, the paternally expressed 10 (*PEG10*) sponging *miR-101-3p*, which targets kinesin family member 2A (*KIF2A*) [76], and SMAD5 antisense RNA 1 (*SMAD5-AS1*) that binds *miR-135b-5bp* to regulate the translation of the APC regulator of the WNT signaling pathway (*APC*) [77].

Small nucleolar RNAs (snoRNAs) are a class of small RNA molecules that assist in chemical modifications of other regulatory RNAs (e.g., rRNAs and tRNAs) [78,79]. Similar to miRNAs, snoRNAs are processed from much longer host genes, which are categorized as lncRNAs. In DLBCL, three snoRNA host genes have been proposed to function as miRNA sponges: small nucleolar RNA host gene 12 (*SNHG12*) sequesters *miR-195* to control the cell growth, migration, and invasion of DLBCL cells in vitro [80]; small nucleolar RNA host gene 14 (SNHG14) sponges *miR-5590-3p* to upregulate Zinc finger E-box binding homeobox 1 (*ZEB1*); and small nucleolar RNA host gene 16 (*SNHG16*) sequesters *miR-497-5p*, to derepress the Pim-1 proto-oncogene, serine/threonine kinase (*PIM1*).

Since a single miRNA can modulate hundreds of mRNAs, it is not surprising that one miRNA can potentially be sequestered by several lncRNAs. *MiR-195*, which is a member of the *miR-15/107* family, is a known tumor suppressor, whose dysregulation is linked to human diseases, including Alzheimer’s disease, cardiac hypertrophy, and many cancers [81,82]. As mentioned above [80], *miR-195* is sequestered by *SNHG12*. In DLBCL, another lncRNA, *MALAT1,* was shown to regulate expression of the CD274 molecule (*PD-L1*) via *miR-195* [83]. If the search is extended to outside of DLBCL, more than two dozen lncRNAs are suggested to sponge *miR-195*, including AFAP1 antisense RNA 1 (*AFAP1-AS1*) [84], AGAP2 antisense RNA 1 (*AGAP2-AS1*) [85], cytoskeleton regulator RNA (*CYTOR*) [86], maternally expressed 3 (*MEG3*) [87], *NEAT1* [88], OIP5 antisense RNA 1 (*OIP5-AS1*) [89], urothelial cancer associated 1 (*UCA1*) [90,91], and *XIST* [92] to name a few (Figure 1D). Thus, it is clear that the studying of lncRNAs as miRNA sponges is far more complex than one would hope for, which suggests that it would be very challenging to use lncRNAs as miRNA sponges for therapeutic purposes.

## 4. Differential Expression of lncRNAs in Rituximab Sensitive and Resistant DLBCL Cell Lines

Understanding the role of lncRNAs in rituximab-resistant DLBCL phenotype could potentially guide the development of improved DLBCL therapies, thereby increase the survival rate of DLBCL patients, and cut the treatment costs. As evident from the current findings of lncRNAs in DLBCL, the number of lncRNAs identified and characterized is still limited. To provide further insights into the role of lncRNAs in DLBCL, we analyzed the transcription profiles of DLBCL cell lines, which are either sensitive or resistant to rituximab, by RNA-sequencing (Figure 2A). These line cells are from different individuals and were not treated with rituximab to uncover the intrinsic transcriptomic differences that underlie drug resistance. We uncovered 195 up- and 428 down-regulated genes in rituximab-resistant DLCBL cell lines compared to sensitive cell lines (Figure 2B), suggesting that there are large gene expression differences inherent to rituximab resistance. When Gene Ontology (GO) analysis for differentially expressed genes (both up- and down-regulated genes combined) was performed, several key signaling pathways known to be involved in tumor development were enriched (Figure 2C); most notably the GTPase signaling pathway [93,94] (Figure 2D), confirming the intrinsic differences in DLBCL cell lines compared.

Of the 458 differentially expressed genes, 123 are classified as lncRNAs by the latest annotation provided by the Ensembl database (GRCh38 version 100), comprising 54 up- and 69-down regulated lncRNAs (Appendix A). To initiate the dissection of the molecular signatures of the differentially expressed lncRNAs, additional bioinformatic analyses were performed. First, the promoter regions of the differentially expressed lncRNAs were examined for potential binding sites of transcription factors to uncover gene regulatory networks of lncRNA expression. Among the 350 transcription factors with potential binding sites in the promoter regions, 10 transcription factors were differentially expressed in rituximab-resistant compared to rituximab-sensitive DLCBL cell lines (*p* < 0.05) (Figure 3A; Appendix A). Among the differentially expressed transcription factors, the homeodomain transcription factor Meis homeobox 1 (MEIS1) is predicted to bind most sites in the highest number of promoter regions of the differentially expressed lncRNAs. Although MEIS1 is down-regulated in rituximab-resistant DLCBL cell lines, a recent study reported that the MYC proto-oncogene, bHLH transcription factor (MYC)-dependent down-regulation of MEIS1 is linked to the tumor development and progression via elevated homeobox B13 (HOXB13) expression and androgen receptor (AR) activity in prostate cancers [97], suggesting that the resistance to rituximab may be linked to the down-regulation of MEIS1, which, in turn, could regulate the expression of downstream lncRNAs. Furthermore, the Kruppel like factor 5 (KLF5) is the only up-regulated transcription factor in rituximab-resistant DLCBL cell lines. Interestingly, a previous report found KLF5 to be upregulated in human breast cancer cells treated with the HER2/epidermal growth factor receptor inhibitor, lapatinib [98]. Together with its family member KLF4, KLF5 was shown to induce expression of the anti-apoptotic factor, MCL1 (a BCL2 family membe*r*), to coordinate the gene regulatory program in resistance to lapatinib. Thus, further studies are warranted to uncover the importance of KLF5 in DLCBL in relation to regulation of lncRNA expression and rituximab resistance.

Besides lncRNAs functioning as miRNA sponges, increasing evidence suggests that many lncRNAs sequester RNA-binding proteins (RBPs) to influence mRNA degradation and stability. Such lncRNAs are collectively called RBP sponges [101]. Interestingly, the up-regulated lncRNAs in rituximab-resistant DLCBL cell lines harbor many putative RBP binding sites within their sequences (Appendix A). In particular, the cholesterol-induced regulator of metabolism RNA (*CHROMR*, also known as *CHROME* and *PRKRA-AS1*) has the fourth highest number of RBP binding sites among all up-regulated lncRNAs examined (Appendix A). A previous study showed that *CHROMR* expression is increased in the plasma and atherosclerotic plaques of individuals with coronary artery disease [102]. Mechanistically, *CHROMR* either sequesters or degrades a set of miRNAs (i.e., *miR-27b*, *miR-33a*, *miR-33b*, and *miR-128*) to repress the expression of genes mediating cholesterol transport in human hepatocytes and macrophages [102]. As cholesterol metabolism has been reported to drive tumor growth and invasion [103], it would be of high interest to study the intertwined link between cholesterol metabolism and DLCBL [104]. Given that many RBPs are involved in cholesterol metabolism and *CHROMR* is 3.4-fold up-regulated in rituximab-resistant DLCBL cell lines [103], in addition to being miRNA sponge, *CHROMR* could function as an RBP sponge to modulate the translation of genes mediating cholesterol transport (Figure 3B).

## 5. Materials and Methods

### 5.1. Cell Culture and Treatment

In this study, the following six human DLBCL-derived cell lines were used: NU-DHL-1, HT, MC-116, SU-DHL-4 (DSMZ, German Collection of Microorganisms and Cell Cultures); OCI-Ly8, and SU-DHL-8 (provided by Dr. Jose A. Martinez-Climent, Molecular Oncology Laboratory, University of Navarra, Pamplona, Spain). The cell lines were cultured under standard conditions at 37 °C in a humidified atmosphere of 95% air and 5% CO2 with RPMI-1640 medium containing 10% fetal bovine serum (FBS) and 1% penicillin/streptomycin (P/S) for no longer than 20 passages. All cell lines were authenticated by DNA barcoding, as previously described [95,96].

### 5.2. RNA Isolation and RNA-seq Assay

Total RNA was extracted using a modified protocol combining TRIzol Reagent (Invitrogen, Paisley, UK) and mirVana miRNA Isolation Kit (Ambion/ThermoFisher Scientific, Grand Island, NY, USA), as previously described [105]. The RNA quality and concentration of each sample were determined by Agilent 2100 Bioanalyzer analysis (Agilent Technologies, Santa Clara, CA, USA) and NanoDrop ND-1000 spectrophotometer (ThermoFisher Scientific), respectively. The total RNA of each sample was sent to AROS Applied Biotechnology AS (Aarhus, Denmark) for poly-A selected, pair-end RNA-seq via Illumina HiSeq 2000 platform. The generated and analyzed data were deposited in the Gene Expression Omnibus (accession ID: GSE159852)

### 5.3. Data Analysis

For RNA-seq data analysis, fastp [106] was used to trim the first seven base pairs, detect paired-end adapters, and analyze overrepresented sequences. After quality control, the trimmed reads were aligned against GRCh38 genome build (version 100 from the Ensembl database) using the Spliced Transcripts Alignment to a Reference (STAR) software [107]. Differential expression analysis was performed using edgeR [108]. The Trimmed Mean of M-values (TMM) method was used to normalize the data to obtain counts per million reads mapped (CPM) values.

A volcano plot was generated using Zenodo [109]. To draw heat maps and run hierarchical clustering, MultiExperiment Viewer (MeV) [110] was used. The bindings of transcription factors were predicted via CiiiDER [111] with its default settings, including the analysis at −1500 bases upstream to +500 bases downstream of the transcription start site (TSS) of each lncRNA. The putative bindings of RBPs were downloaded from the oRNAment database [112]. The network of RBPs was drawn via NetworkAnalyst [100].

## 6. Concluding Remarks

This review summarizes recent findings of lncRNAs in DLBCL. Although several studies of lncRNAs in DLBCL have been published [67,68,69,70,113,114], the number of functionally studied lncRNAs is still limited, which precludes the defined diagnostic and prognostic importance of lncRNAs in DLBCL patients. To this end, we provide molecular signatures of lncRNAs in rituximab-resistant and -sensitive DLCBL cell lines, respectively. A study published in 2017 identifies 17 lncRNAs that can discriminate between two major molecular subtypes of DLBCL, activated B-cell-like (ABC) and germinal center B-cell-like (GCB), with high specificity, which the authors termed these seventeen lncRNAs as SubSigLnc-17. However, in our RNA-seq data, SubSigLnc-17 were not differentially expressed, which promoted us to further analyze the differentially expressed lncRNAs in our RNA-seq data [115]. We report on initial bioinformatic analyses of the differentially expressed lncRNAs; however, further biological validation and functional studies are required to uncover the biological roles of the identified lncRNAs in DLCBL.

## Figures and Tables

**Figure 1 ncrna-07-00001-f001:**
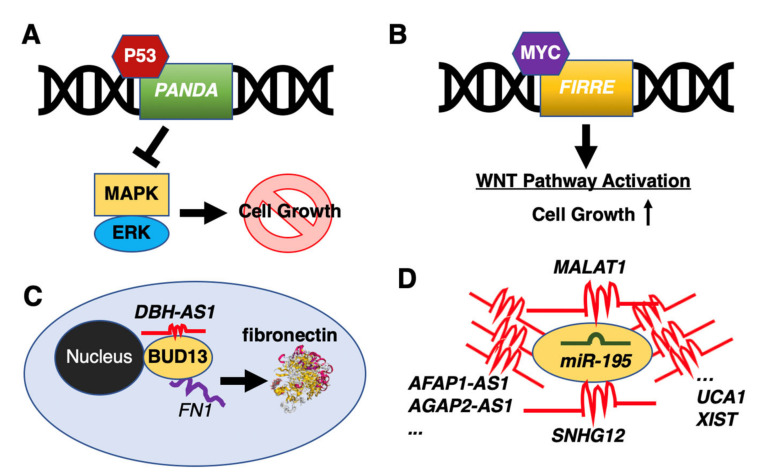
Representative long non-coding RNAs (lncRNAs) in diffuse large B-cell lymphoma (DLBCL). (**A**) The p53-activated *PANDA* inhibits cell growth through inactivation of the MAPK/ERK pathway. (**B**) The MYC-activated *FIRRE* activates the Wnt/β-catenin signaling pathway to facilitate DLBCL cell growth. (**C**) The lncRNA *DBH-AS1* controls the translation of fibronectin 1 (FN1) through binding to the RNA-binding protein, BUD13. (**D**) Many lncRNAs have been suggested to function as miRNA sponges to sequester *miR-195*.

**Figure 2 ncrna-07-00001-f002:**
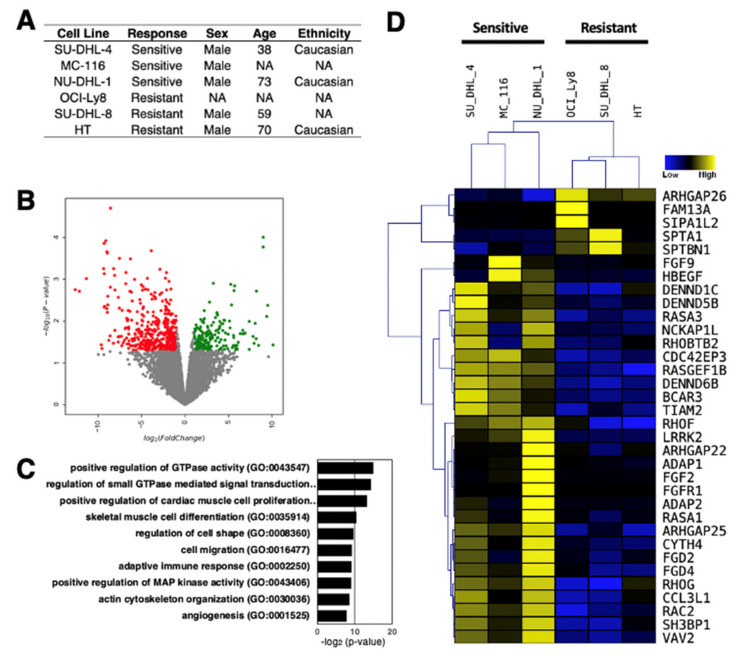
RNA-seq analysis of three rituximab-resistant and three rituximab-sensitive DLBCL cell lines. (**A**) Sample information for the DLBCL cell lines used in this study [95,96]. (**B**) Volcano plot of RNA-seq data. With the threshold values of 2-fold and *p* < 0.05, there are 195 up- and 428 down-regulated genes in rituximab-resistant compared to rituximab-sensitive DLCBL cell lines (n = 3 biologically independent samples). (**C**) Gene ontology (GO) of differentially expressed genes (both up- and down-regulated genes combined). (**D**) Hierarchical clustering of differentially expressed genes involved in GTPase signaling pathway.

**Figure 3 ncrna-07-00001-f003:**
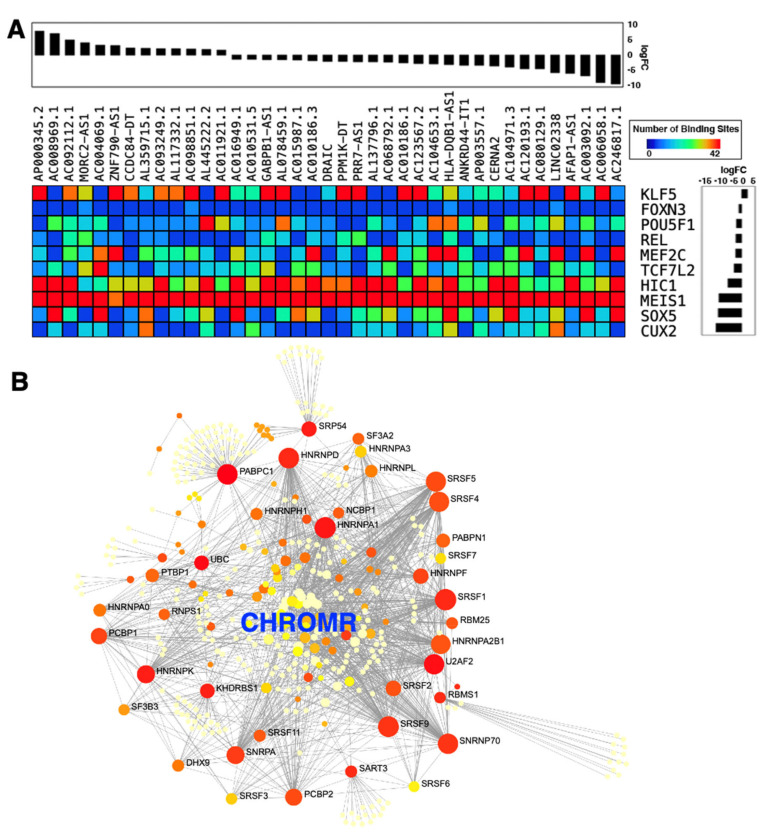
Molecular characteristics of differentially expressed lncRNAs in rituximab-resistant DLBCL cell lines compared to rituximab-sensitive cell lines. (**A**) Enrichment of transcription factor binding in the promoter regions of differentially expressed lncRNAs. The following 10 transcription factors with the highest number of binding sites in these promoter regions are shown in the image along with the log-transformed fold change (logFC) values in rituximab-resistant compared to rituximab-sensitive DLCBL cell lines: cut like homeobox 2 (CUX2); forkhead box N3 (FOXN3); HIC ZBTB transcriptional repressor 1 (HIC1); Kruppel like factor 5 (KLF5); myocyte enhancer factor 2C (MEF2C); Meis homeobox 1 (MEIS1); POU class 5 homeobox 1 (POU5F1); REL proto-oncogene, NF-kB subunit (REL); SRY-box transcription factor 5 (SOX5); and transcription factor 7 like 2 (TCF7L2). To allow for the visual inspection, 39 of 123 differentially expressed lncRNAs are shown along with the logFC values in rituximab-resistant compared to -sensitive DLCBL cell lines. (**B**) Protein-protein interactions (PPIs) of RNA-binding proteins (RBPs) predicted to bind CHROMR. PPIs are based on the information provided by the STRING database [99] visualized through NetworkAnalyst [100].

## Data Availability

The data sets analyzed in this study can be found in the Zenodo repository (https://doi.org/10.5281/zenodo.4146742) and in the Gene Expression Omnibus repository (https://www.ncbi.nlm.nih.gov/geo/) with the reference number GSE159852.

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
