# Peer review of "Long Non-Coding RNAs in Diffuse Large B-Cell Lymphoma"

_ncrna, 2020, doi:10.3390/ncrna7010001_

Round 1

Reviewer 1 Report

The manuscript of Karstensen et al. is a hybrid paper providing an overview of lncRNAs in DLBCL and an analysis of differentially expressed lncRNAs in cell lines that are either resistant or susceptible to treatment with rituximab. The authors should address the following to be suitable for publication.

  1. The review component is missing key references. Notably, papers such as Verma et al., Genome Med. 2015;7(1):110; Zhou et al. Mol Cancer. 2017;16(1):16; Sun et al. Sci Rep. 2016;6(1):27842 should be included.
  2. Tying in with the above references, there is no discussion of the relationship of lncRNAs to ABC/GCB classification, although this is widely used to stratify patients and the two groups have different underlying biology. The authors should also indicate whether the cell lines used belong to one or the other category.
  3. Given that rituximab has multiple mechanisms of action, which include the level of cell-surface CD20 and non-direct effects such as complement activation and ADCC, it would potentially be helpful to specify that the described analysis would likely apply to a subset of resistant patients.
  4. The authors have not provided adequate details about the nature of the rituximab-resistant cell lines. There is no reference listed in the initial description in section 4, lines 151-159, although one assumes that it is based on the authors’ previous publication (reference 103, Due et al.). However, in the previous paper neither the rituximab sensitivity for HT or MC-116 were reported and there appeared to be a mixed phenotype  for NU-DHL-1, dependent on the cell seeding concentration/dose of rituximab.
  5. Figure 2 illustrates that the most enriched GO term associated with differentially expressed genes is regulation of GTPase activity. Are any of the genes listed in part 2 known to be regulated by lncRNAs?
  6. As shown in Fig 3A, The presence of the MEIS1 binding motif appears to be evenly distributed amongst the differentially expressed lncRNA promoters at a very high density (approx. 40 sites in a 2kb region). Is this unique to the 39 loci shown? How does this compare to a similar sized group of lncRNAs that are expressed at the same level in the two cell groups?
  7. Are the PPIs shown in Fig 3B based on known direct physical interactions or are these functional associations?

Author Response

The manuscript of Karstensen et al. is a hybrid paper providing an overview of lncRNAs in DLBCL and an analysis of differentially expressed lncRNAs in cell lines that are either resistant or susceptible to treatment with rituximab. The authors should address the following to be suitable for publication.

The review component is missing key references. Notably, papers such as Verma et al., Genome Med. 2015;7(1):110; Zhou et al. Mol Cancer. 2017;16(1):16; Sun et al. Sci Rep. 2016;6(1):27842 should be included.

Response: Thank you very much for carefully reading our manuscript. Zhou et al. Mol Cancer. 2017 [PMID: 28245823] is about pancreatic cancer so we could not add this reference to our manuscript. However, we now cite the other two articles in the Concluding remarks section as follows:

“This review summarizes recent findings of lncRNAs in DLBCL. Although several studies of lncRNAs in DLBCL have been published [67-70,113,114], the number of functionally studied lncRNAs is still limited, which precludes the defined diagnostic and prognostic importance of lncRNAs in DLBCL patients.”

References

  1. Sun, J.; Cheng, L.; Shi, H.; Zhang, Z.; Zhao, H.; Wang, Z.; Zhou, M. A potential panel of six-long non-coding RNA signature to improve survival prediction of diffuse large-B-cell lymphoma. Sci Rep 2016, 6, 27842, doi:10.1038/srep27842.

  1. Verma, A.; Jiang, Y.; Du, W.; Fairchild, L.; Melnick, A.; Elemento, O. Transcriptome sequencing reveals thousands of novel long non-coding RNAs in B cell lymphoma. Genome Med 2015, 7, 110, doi:10.1186/s13073-015-0230-7.

Tying in with the above references, there is no discussion of the relationship of lncRNAs to ABC/GCB classification, although this is widely used to stratify patients and the two groups have different underlying biology. The authors should also indicate whether the cell lines used belong to one or the other category.

Response: A study published in 2017 [Zhou M, Zhao H, Xu W, Bao S, Cheng L, Sun J (2017) Discovery and validation of immune-associated long non-coding RNA biomarkers associated with clinically molecular subtype and prognosis in diffuse large B cell lymphoma. Mol Cancer. 2017 Jan 19;16(1):16. PMID: 28103885] identifies 17 lncRNAs that can discriminate between the ABC and GCB subtypes of DLBCL. These seventeen lncRNAs were identified as candidate biomarkers and integrated into form a lncRNA-based signature (termed SubSigLnc-17), which was able to discriminate between GCB and ABC subtypes with AUC of 0.974, specificity of 89.6% and sensitivity of 92.5%. These 17 lncRNAs are not differentially expressed in rituximab-resistant DLCBL cell lines compared to sensitive cell lines in our RNA-seq data. Thus, based on the differentially expressed lncRNAs, the ABC and GCB subtypes of DLBCL is not an obvious explanation of the differentially expressed lncRNAs in our RNA-seq data. Additionally, rituximab is equally beneficial in both ABC and GCB subtypes of DLBCL; supporting that molecular subtyping of DLBCL into ABC/GCB is not a major determinant of sensitivity towards rituximab and has therefore been omitted as a parameter in this manuscript.

Given that rituximab has multiple mechanisms of action, which include the level of cell-surface CD20 and non-direct effects such as complement activation and ADCC, it would potentially be helpful to specify that the described analysis would likely apply to a subset of resistant patients.

The authors have not provided adequate details about the nature of the rituximab-resistant cell lines. There is no reference listed in the initial description in section 4, lines 151-159, although one assumes that it is based on the authors’ previous publication (reference 103, Due et al.). However, in the previous paper neither the rituximab sensitivity for HT or MC-116 were reported and there appeared to be a mixed phenotype for NU-DHL-1, dependent on the cell seeding concentration/dose of rituximab.

Response: We apologize for not providing detailed information about the DLBCL cell lines. These cell lines were derived from different individuals and NOT treated with rituximab. In this manuscript, we provide the intrinsic nature of transcriptome differences in rituximab-resistant and -sensitive cell lines. We added the following sentence to the Results section for clarification:

“These cell lines are from different individuals and were not treated with rituximab to uncover the intrinsic transcriptomic differences that underlie drug resistance.”

Figure 2 illustrates that the most enriched GO term associated with differentially expressed genes is regulation of GTPase activity. Are any of the genes listed in part 2 known to be regulated by lncRNAs?

Response: There are 30 differentially expressed genes categorized under positive regulation of GTPase activity (GO:0043547). When each of these genes was searched via the PubMed database (accessed on December 11, 2020) for their relationships to lncRNAs, there are several articles reporting lncRNAs regulating the genes involved in GTPase activity, including the lncRNA NEAT1 sponging miR-1277-5p, which targets ARHGAP26 [PMID: 33069733]; the host gene, MIR31HG (also known as LOC554202) regulating RASA1 via miR-31 [PMID: 31762810]; and the lncRNA PCGEM1 sponging miR-433-3p to regulate FGF2 expression [PMID: 31958075]. However, none of these lncRNAs are differentially expressed in rituximab-resistant DLCBL cell lines compared to sensitive cell lines in our RNA-seq data (the list of differentially expressed lncRNAs are provided in Supplementary Table S1), suggesting that more research is needed to uncover the possible relationships between the differentially expressed lncRNAs identified in our RNA-seq data and GTPase activity.

As shown in Fig 3A, The presence of the MEIS1 binding motif appears to be evenly distributed amongst the differentially expressed lncRNA promoters at a very high density (approx. 40 sites in a 2kb region). Is this unique to the 39 loci shown? How does this compare to a similar sized group of lncRNAs that are expressed at the same level in the two cell groups?

Response: To identify the binding sites of transcriptional regulators, we used the CiiiDER tool with its default settings. Of 123 differentially expressed lncRNAs from our RNA-seq data, CiiiDER identified statistically significant enrichment of binding of 17 transcription factors to the promoters of 112 lncRNAs. These data are provided as Supplementary Table S2. As stated in the main text, MEIS1 binds the highest number of promoters; not only those shown in Figure 3A.

Are the PPIs shown in Fig 3B based on known direct physical interactions or are these functional associations?

Response: The PPIs were drawn using the NetworkAnalyst web tool by restriction to the information provided by the STRING database. The STRING database provides PPIs based on the experimental results as well as those extracted from text mining of published articles and co-expression.

Reviewer 2 Report

The author has provided the lncRNA role in DLBCL, the review is good and adds value to what is already, however, I would like them to add on the details on the Antisense RNA in their review, for example, FAS-AS1 has been shown in the DLBCL and all other lymphomas to be a critical determinant of the lymphoma survival. also, the authors mention that the cell lines they selected are resistant or sensitive to rituximab, however, they are not syngenic, and hence the drug resistance could be because they are inherently or genetically different I would like them to discuss the lncRNA or GEP from the syngenic background. 

Author Response

The author has provided the lncRNA role in DLBCL, the review is good and adds value to what is already, however, I would like them to add on the details on the Antisense RNA in their review, for example, FAS-AS1 has been shown in the DLBCL and all other lymphomas to be a critical determinant of the lymphoma survival.

Response: Thank you very much for your positive response. We added the following sentence to address the importance of antisense RNAs:

“Another example is the EZH2-mediated lncRNA, FAS antisense RNA 1 (FAS-AS1), which binds to the RNA-binding protein, RNA binding motif protein 5 (RBM5), to control alternative splicing of Fas cell surface death receptor (FAS) mRNA [73].”

also, the authors mention that the cell lines they selected are resistant or sensitive to rituximab, however, they are not syngenic, and hence the drug resistance could be because they are inherently or genetically different I would like them to discuss the lncRNA or GEP from the syngenic background.

Response: We apologize for not being clear on the RNA-seq data provided. These cell lines were derived from different individuals (thus, non-syngenic) and NOT treated with rituximab. To make this point, we provide the intrinsic nature of transcriptome differences in rituximab-resistant and -sensitive cell lines. We added the following sentence to the Results section:

“These cell lines are from different individuals and were not treated with rituximab to uncover the intrinsic transcriptomic differences that underlie drug resistance.”

Round 2

Reviewer 1 Report

There are some points that the authors misunderstood that need to be resolved.

  1. There is some confusion about the paper by Zhou et al., 2017. The one I listed is Zhou et al. Mol Cancer. 2017;16(1):16, which is the same one that the authors use to rebut the point about ABC vs GCB and would be a valuable paper to include.

The Zhou paper about pancreatic cancer is 16(1):52.

I think it would be worth including a statement in the text summarising the information included in the rebuttal about the fact that the any rituximab-related lcRNAs do not depend on the ABC/GCB classification.

  1. It was clearly described that the cell lines used for analysis were not treated with rituximab. That was not the problem.

The issue was that there should be some previous paper or included experiment to provide evidence that the cells lines used were either sensitive or resistant. How do we know which ones fall into each category?

Author Response

There are some points that the authors misunderstood that need to be resolved.

There is some confusion about the paper by Zhou et al., 2017. The one I listed is Zhou et al. Mol Cancer. 2017;16(1):16, which is the same one that the authors use to rebut the point about ABC vs GCB and would be a valuable paper to include.

The Zhou paper about pancreatic cancer is 16(1):52.

I think it would be worth including a statement in the text summarising the information included in the rebuttal about the fact that the any rituximab-related lcRNAs do not depend on the ABC/GCB classification.

Answer: Thank you very much for further clarifying your comments. We now added the above mention article as follow:

A study published in 2017 identifies 17 lncRNAs that can discriminate between two major molecular subtypes of DLBCL, activated B-cell-like (ABC) and germinal center B-cell-like (GCB), with high specificity, which the authors termed these seventeen lncRNAs as SubSigLnc-17. However, in our RNA-seq data, SubSigLnc-17 were not differentially expressed, which promoted us to further analyze the differentially expressed lncRNAs in our RNA-seq data [115].

References

  1. Zhou, M.; Zhao, H.; Xu, W.; Bao, S.; Cheng, L.; Sun, J. Discovery and validation of immune-associated long non-coding RNA biomarkers associated with clinically molecular subtype and prognosis in diffuse large B cell lymphoma. Mol Cancer 2017, 16, 16, doi:10.1186/s12943-017-0580-4.

It was clearly described that the cell lines used for analysis were not treated with rituximab. That was not the problem.

The issue was that there should be some previous paper or included experiment to provide evidence that the cells lines used were either sensitive or resistant. How do we know which ones fall into each category?

Answer: The requested information is provided as an embedded table in Figure 2A. To further clarify the origins of these cell lines, we now modify the corresponding figure legend as follow:

Figure 2. RNA-seq analysis of three rituximab-resistant and three rituximab-sensitive DLBCL cell lines. (A) Sample information for the DLBCL cell lines used in this study [95,96].

References

  1. Laursen, M.B.; Reinholdt, L.; Schonherz, A.A.; Due, H.; Jespersen, D.S.; Grubach, L.; Ettrup, M.S.; Roge, R.; Falgreen, S.; Sorensen, S., et al. High CXCR4 expression impairs rituximab response and the prognosis of R-CHOP-treated diffuse large B-cell lymphoma patients. Oncotarget 2019, 10, 717-731, doi:10.18632/oncotarget.26588.

  1. Due, H.; Brondum, R.F.; Young, K.H.; Bogsted, M.; Dybkaer, K. MicroRNAs associated to single drug components of R-CHOP identifies diffuse large B-cell lymphoma patients with poor outcome and adds prognostic value to the international prognostic index. BMC Cancer 2020, 20, 237, doi:10.1186/s12885-020-6643-8.

Reviewer 2 Report

THEY HAVE REPOSNDED TO MY COMMENTS

Author Response

Thank you very much.